# *Bartonella quintana* Infection in Canada: A Retrospective Laboratory Study and Systematic Review of the Literature

**DOI:** 10.3390/pathogens13121071

**Published:** 2024-12-06

**Authors:** Carl Boodman, Leslie R. Lindsay, Antonia Dibernardo, Courtney Loomer, Yoav Keynan, Matthew P. Cheng, Cédric P. Yansouni, Nitin Gupta, Heather Coatsworth

**Affiliations:** 1Division of Infectious Diseases, Department of Internal Medicine, University of Manitoba, Winnipeg, MB R3E 0T6, Canada; yoav.keynan@umanitoba.ca; 2Unit of Neglected Tropical Diseases, Department of Clinical Sciences, Institute of Tropical Medicine, Nationalestraat 155, 2000 Antwerp, Belgium; 3Department of Medical Sciences, University of Antwerp, 2610 Antwerpen, Belgium; nityanitingupta@gmail.com; 4Public Health Agency of Canada, Winnipeg, MB R3E 3R2, Canada; robbin.lindsay@phac-aspc.gc.ca (L.R.L.); antonia.dibernardo@phac-aspc.gc.ca (A.D.); courtney.loomer@phac-aspc.gc.ca (C.L.); heather.coatsworth@phac-aspc.gc.ca (H.C.); 5Divisions of Infectious Diseases and Medical Microbiology, McGill University Health Centre, Montréal, QC H4A 3J1, Canada; matthew.cheng@mcgill.ca (M.P.C.); cedric.yansouni@mcgill.ca (C.P.Y.); 6JD MacLean Centre for Tropical and Geographic Medicine, McGill University, Montréal, QC H3A 1A1, Canada; 7Department of Infectious Disease, Kasturba Medical College, Manipal, Manipal Academy of Higher Education, Manipal 576104, India

**Keywords:** trench fever, endocarditis, pediculosis, ectoparasitosis, homelessness

## Abstract

**Background:***Bartonella quintana* is a body-louse-borne bacterium. Canadian *B. quintana* disease has been reported primarily in populations experiencing homelessness and in Indigenous communities with limited access to water. We sought to understand the epidemiology of *B. quintana* in Canada. **Methods:** This study combined an analysis of laboratory data from Canada’s National Microbiology Laboratory (NML) with a systematic review of the literature. Laboratory data included quantitative polymerase chain reaction (qPCR) cycle threshold values and indirect immunofluorescent antibody titers with the year and province of the sample acquisition. For the systematic review, we searched PubMed, Scopus, Embase, and Web of Science for articles published before 15 July 2024, with terms related to *B. quintana* in Canada. **Results:** Thirty-three individuals with qPCR-positive *B. quintana* were documented in seven provinces and one territory. The number of cases increased over time (*p*-value = 0.005), with the greatest number of cases being reported in 2022 and 2023. The percent positivity for the *B. quintana* qPCR performed at the NML increased over time (*p*-value = 0.036). The median immunoglobulin G titer demonstrated a sustained increase starting in 2017. The systematic review identified fourteen individuals with qPCR-positive *B. quintana* (none had a qPCR performed at the NML) and seven probable cases of *B. quintana* disease. Four of these twenty-one individuals from the systematic review died (19%). All fatalities were attributed to endocarditis. **Conclusions**: The detection of *B. quintana* disease in seven provinces and one territory suggests that *B. quintana* has a national distribution. *B. quintana* disease is increasingly diagnosed in Canada, indicating ongoing transmission across geographic settings.

## 1. Introduction

*Bartonella quintana* is an intracellular Gram-negative bacillus transmitted by infected body lice (*Pediculus humanus corporis*) feces [1]. The bacterium was discovered in 1915 as the cause of trench fever, a relapsing febrile illness affecting World War I soldiers. *B. quintana* was later determined to cause chronic bacteremia, bacillary angiomatosis, and infective endocarditis [2,3,4,5]. The bacterium enters the human bloodstream when individuals with a body lice infestation experience the contamination of skin abrasions with body lice feces containing *B. quintana* [1,6]. Due to the bacterium’s intracellular niche and median replication time of 3 h, *Bartonella* species cannot be identified by a regular blood culture with a five-day incubation and are characterized as culture-negative bacterial pathogens [7,8,9]. The 2023 guideline updates to the modified Duke criteria added *Bartonella* molecular detection and serologic positivity with elevated titers as major diagnostic criteria for infective endocarditis [8].

As body louse infestations occur in contexts where there is inadequate access to water to maintain personal hygiene, *B. quintana* infection is associated with homelessness, over-crowding, and armed conflict [10,11,12,13,14]. Recent outbreaks of *B. quintana* have occurred among people experiencing homelessness in high-income countries, refugees and internally displaced populations in low-income countries, and Canadian indigenous communities with limited access to running water [1,11,12,15,16].

*B. quintana* disease has been acquired in 40 countries, on all continents except Antarctica, in both urban and rural settings [11]. Epidemiologic patterns differ between high-income countries (HICs), and low- and middle-income countries (LMICs) [11]. In HICs, *B. quintana* infection is predominantly associated with urban homelessness, the male gender, and substance use disorder [1,17,18,19]. These risk factors were identified in some of the earliest descriptions of *B. quintana* in the 1990s, and have been corroborated by recent studies [18,19]. In a 2022 study from Denver, Colorado, 93% of *B. quintana* cases were male, 79% experienced homelessness, and 79% had a substance use disorder [18]. In LMICs, *B. quintana* also causes disease among housed individuals, including females and children, living in rural conditions where access to water is limited [11]. Arthropod studies provide an additional window into *B. quintana* epidemiology. *B. quintana* DNA has been detected in lice from most continents [20]. The elevated number of *B. quintana*-DNA-positive lice from certain countries in Asia and Africa suggests an undiagnosed burden in LMICs [20].

In Canada, the first description of *B. quintana* disease was published in 1996, describing a fatal case of aortic valve endocarditis [21]. From 1996 to 2020, seven additional Canadian cases of *B. quintana* were published, including three individuals with endocarditis, three with bacteremia, and one with bacillary angiomatosis [15,22,23,24,25]. In 2020, Canada’s largest outbreak of *B. quintana* was described among four individuals with endocarditis in Winnipeg, Manitoba [26]. This cluster prompted a retrospective investigation revealing eleven prior cases of *B. quintana* in Manitoba, including two fatalities attributed to endocarditis [27]. In 2022, an 11-year-old child from a remote indigenous community in Northern Manitoba required valve replacement surgery for *B. quintana* endocarditis, the first childhood case to be acquired in a high-income country [12].

Serologic and molecular testing for *B. quintana* is available at the National Microbiology Laboratory (NML, Public Health Agency of Canada) in Winnipeg, Manitoba, and through individual provinces. However, *B. quintana* is not a nationally notifiable disease, and its epidemiology in Canada remains largely undescribed [28]. This study aims to comprehensively describe the *B. quintana* epidemiology in Canada by combining laboratory data from the main reference laboratory providing testing nationally with a systematic review of publications describing Canadian cases. The systematic review includes clinical and epidemiologic details absent from the NML data, enabling a more detailed description of clinical and epidemiologic trends.

## 2. Materials and Methods

### 2.1. Study Design

This study combines a retrospective analysis of laboratory data from the NML with a systematic review of the published literature. Our study questions are as follows: What is the epidemiology of *B. quintana* in Canada and how do the results of Canadian *B. quintana* testing change over time?

### 2.2. Laboratory Data

The NML performed molecular and serologic testing for *B. quintana.* Molecular testing involved nucleic acid sample extraction via a DNeasy or DNAmini kit (Qiagen, Hilden, Germany) and screening for the *Bartonella* genus using quantitative polymerase chain reaction (qPCR) of the ITS3 gene. Samples that were screened as ITS3-positive were further tested to determine the infective *Bartonella* species, using the *yopP* gene for *B. quintana* and the *pap31* gene for *B. henselae*. Samples with cycle threshold (Ct) values of <40 for both ITS3 and *yopP* were considered positive for *B. quintana*. This study used molecular data from 1 January 2017, until 31 December 2023.

Serologic testing was performed using a commercial indirect immunofluorescent antibody assay (IFA) (DiaSorin, Mississauga, ON, Canada) that detects immunoglobulin G (IgG) to *B. quintana* and *B. henselae* antigens [29]. A positive IFA result was defined as a single titer of 1:64 or greater to the *B. quintana* antigen, as suggested by the IFA package insert, although titers greater or equal to 1:256 are more suggestive of active disease [29]. A four-fold or greater increase in IgG titers between acute and convalescent samples drawn at least two weeks apart indicates a current infection. Titers greater or equal to 1:1024 (or 1:800 using other immunofluorescence antibody assays) are cut-offs proposed for the diagnosis of infective endocarditis [8]. Serologic data were available from 1 January 2008, until 31 December 2023.

### 2.3. Systematic Literature Search Strategy

A systematic review was performed to identify published cases excluded from the NML data. We searched the PubMed, Scopus, Embase, and Web of Science databases from 1 January 1915 (the year of *B. quintana* discovery) to 15 July 2024, using the following search string, with associated MeSH terms and Boolean operators: {(*Bartonella quintana* OR *Rochalimaea quintana* OR *Rickettsia quintana* OR Trench fever) AND (Canada)}. *Rochalimaea quintana* and *Rickettsia quintana* were included as they are previous scientific names for *B. quintana.* Additionally, we searched reference lists of selected publications to identify other reported cases not published elsewhere. There was no language restriction, although searches were performed in English. This review followed the Preferred Reporting Items for Systematic Reviews and Meta-Analyses (PRISMA) guidelines for performing systematic literature reviews and was registered in the International Prospective Register of Systematic Reviews (PROSPERO; identifier CRD42024569779) [30,31].

### 2.4. Study Selection and Case Definitions

Reports of human clinical cases of *B. quintana* diagnosed in Canada were included. Cases were classified as confirmed or probable *B. quintana* disease based on definitions proposed for similar infections of public health relevance [32]. Confirmed cases identified *B. quintana* to species level by qPCR. Probable *B. quintana* cases demonstrated *B. quintana* positivity on a serological platform, and had a clinical syndrome compatible with *B. quintana* disease (e.g., endocarditis, and bacillary angiomatosis), epidemiologic risk factors consistent with *B. quintana* acquisition (e.g., homelessness, and body lice infestation), and the absence of risk factors for *B. henselae* (e.g., reported cat exposure) [27]. Studies that reported cases of *Bartonella* infection but lacked pathogen identification at the species level and also lacked epidemiologic information were excluded. In vitro studies and review articles without new clinical case information were excluded. Retrospective studies describing previously published data were consolidated with the first report to avoid duplication.

### 2.5. Article Review

Article title and abstract were screened by two individuals (C.B. and N.G.) to determine eligibility for full-text review. Full texts of the articles included after title/abstract screening were reviewed by two independent reviewers (C.B. and N.G.) and reviewer discrepancies were resolved by discussion or with a third author when necessary.

### 2.6. Quality Assessment for Included Studies

Two reviewers (C.B. and N.G.) assessed articles for quality using the JBI critical appraisal checklist for methodological quality and potential bias (Appendix A) [33]. Studies that failed to meet most JBI criteria were excluded from the primary analysis.

### 2.7. Data Extraction from the Systematic Review

Data were manually extracted from included articles by one author (C.B.) using Microsoft Excel (2019, version 16.72) and corroborated by a second author (N.G.) who indicated inconsistencies. The latter were resolved through discussion. For each included reference, we extracted publication-related, epidemiologic, clinical, and diagnostic data (Appendix A).

### 2.8. Statistical Analysis

Descriptive statistics, Mann–Whitney U tests, and Mann–Kendall trend tests were performed post hoc using R version 4.2.2 software (R Foundation for Statistical Computing, 31 October 2022). The Mann–Whitney U test was used to compare groups of Ct values (e.g., blood vs. solid organ tissue) and groups of IFA titers (e.g., titer to *B. quintana* antigen vs. *B. henselae* antigen). The Mann–Kendall trend test was used as a non-parametric test to assess the presence of a monotonic temporal trend. Percent positivity of qPCR was calculated by dividing the number of samples testing positive for *B. quintana* DNA by qPCR per year by the total number of samples processed by the NML for *Bartonella* qPCR per year. Values of *p* < 0.05 were considered significant.

### 2.9. Ethics Approval

This study used published and anonymized data without any personal identifiable information. The Government of Canada Panel on Research Ethics does not require approval for this type of study as it “relies exclusively on secondary use of anonymous information” [34].

## 3. Results

### 3.1. Molecular Testing

A total of 1841 tests for *Bartonella* qPCR were performed at the NML between 2017 and 2023, of which 26 (1.41%, 95% CI [0.87–1.95%]) were positive for *B. quintana* (Table 1). These 26 samples were linked to 19 cases of *B. quintana* disease, as seven cases had multiple sample types submitted (e.g., blood and cardiac valve specimens for individuals with endocarditis). Of these 19 cases, seven were from Alberta, seven were from Ontario, four were from British Columbia, and one was from Quebec (Table 2). Positive *B. quintana* samples increased from 2017 to 2023 (tau = 0.751, *p*-value = 0.031). From 2017–2021, the NML reported zero to two qPCR-positive samples per year. In 2022 and 2023, eight and twelve samples with *B. quintana* DNA were reported each year, respectively. The percent positivity for *B. quintana* qPCR increased over time (tau = 0.714, *p*-value = 0.0355).

Of the 26 positive *B. quintana* samples, 20 were tissue samples (14 cardiac valves from individuals with endocarditis and 6 skin tissues from individuals with bacillary angiomatosis), 5 were whole blood samples, and 1 was a serum sample (Table 2). Tissue samples yielded lower Ct values than blood/serum samples for both the ITS3 and *yopP* gene targets (ITS3: *z*-score = −3.25591, *p*-value = 0.00112. *yopP*: *z*-score = −3.31676, *p*-value = 0.0009). The mean Ct values from cardiac valves were 21.8 for the ITS3 gene and 19.2 for *yopP*. The mean Ct values from skin samples were 26.6 and 23.9, for ITS3 and *yopP*, respectively. Blood and serum samples had mean Ct values of 37.7 and 35.7, for ITS3 and *yopP*, respectively. Four individuals had paired blood and tissue specimens positive for *B. quintana*; all had lower Ct values on tissue than blood (Table 2 and Appendix A).

### 3.2. Serologic Testing

In total, 20504 *Bartonella* IFA tests were processed from 2008 to 2023, of which 962 (4.7%, 95% CI [4.41–4.99%]) demonstrated a positive IgG response to the *B. quintana* antigen (Table 3). While the serologic positivity rate remained stable over time (tau = −0.177, *p*-value = 0.36642), the median IFA titer demonstrated a sustained increase starting in 2017. Provinces that submitted sera with IFA titers greater or equal to 1:1024 (suggestive of infective endocarditis) included Manitoba (37 specimens), Alberta (13), Ontario (13), New Brunswick (12), Nova Scotia (7), Saskatchewan (2), and BC (1). All 11 specimens with titers greater or equal to 1:8192 (the maximum dilution reported by the NML) were submitted from Ontario or Manitoba.

### 3.3. Paired Molecular and Serologic Testing

Eleven qPCR-positive samples had paired IFA results (Table 4). IFA titers were similar for both *B. quintana* and *B. henselae* antigens (*z*-score = −0.39399, *p*-value = 0.69654). IgG titers were identical for *B. quintana* and *B. henselae* antigens in 9 out of 11 samples, and 2 samples had four-fold or greater titers to *B. henselae* (Table 4). Three cases of molecularly confirmed *B. quintana* disease had negative IFA results: all were cases of bacillary angiomatosis with *B. quintana* detected by qPCR on skin tissue. One of the bacillary angiomatosis cases with a negative serology was associated with *B. quintana* DNA detection on a paired blood sample. Among all paired molecular and serologic samples, median IFA titers associated with *B. quintana*-positive cardiac tissue, skin tissue, and blood/serum samples were 1:3072, 1:160, and 1:64, respectively.

### 3.4. Results from Systematic Review

We identified 66 publications through the database search and 1 additional publication from the citation search (Figure 1). After removing 29 duplicates and reviewing titles/abstracts and full texts, 11 publications met the inclusion criteria; 27 publications were excluded as they were in vitro studies without any clinical case information (6), non-human studies (9), review articles (5), did not occur in Canada (5), or did not involve *B. quintana* (2). Apart from one publication in 1996, the ten other included reports were published after 2000. The 11 included publications described 21 cases of *B. quintana* of which 14 were confirmed by qPCR and 7 were probable cases (Appendix A). None of the published cases overlapped with the NML molecular data presented herein. All 14 of the molecularly confirmed cases were diagnosed by 16S rRNA, *rpoB* or *ribC* PCR, and gene sequencing performed at an academic hospital laboratory. Of the 21 cases, 14 were cases of infective endocarditis, 5 were cases of febrile illness, 1 was a case of bacillary angiomatosis, and 1 was a case of leg pain associated with body lice infestation. Embolization was described in 7 of the 14 cases of endocarditis, most commonly to the brain (5) and spleen (2). Four cases died, generating a mortality rate of 19% [21,27,35]. All fatalities were associated with infective endocarditis. Two of the four fatal cases of endocarditis did not undergo valvular replacement surgery compared to two of the ten endocarditis cases that survived. Thirteen of the twenty-one cases had a history of homelessness. Of the eight cases not associated with homelessness, six were acquired in rural areas of Alberta, Manitoba and Québec, one was acquired in Halifax and one was likely acquired in Eritrea prior to immigrating to Manitoba (Appendix A).

### 3.5. Quality Assessment of the Systematic Review

A quality assessment using the JBI critical appraisal checklist for case reports and case series studies revealed that 11 publications had sufficient information to be included in the full-text analysis (Appendix A) [33].

### 3.6. Total of Molecularly Confirmed Cases

Combining the NML data with published reports from the systematic review, 33 molecularly confirmed cases of *B. quintana* have occurred across seven Canadian provinces and one territory. The number of confirmed cases has increased over time (tau = 0.434, *p*-value = 0.005), with the greatest number of cases being reported in 2022 and 2023 (Figure 2). The provinces with the most confirmed cases were Alberta, Ontario, and Manitoba, with nine, seven, and five cases, respectively (Figure 3). Of the 33 PCR-confirmed cases of *B. quintana*, there were 25 cases of infective endocarditis (25/33 = 75.8%).

## 4. Discussion

The detection of *B. quintana* disease from seven provinces and one territory suggests that *B. quintana* is a vector-borne disease with a national distribution, infecting individuals both in urban centers and remote rural communities. Humans are the primary reservoir of *B. quintana* [1]. Thus, each individual case of *B. quintana* disease indicates transmission in the surrounding community. None of the publications from the systematic review described screening close contacts of index *B. quintana* cases, suggesting that additional *B. quintana* cases may be undescribed in many areas [24]. The unrecognized burden of *B. quintana* was demonstrated by a 2024 entomologic study in Winnipeg, Manitoba and a 2022–2023 outbreak among organ transplant recipients in Edmonton, Alberta [17,37,38]. Among 7 Winnipeggers with a body lice infestation, one individual had 218 body lice positive for *B. quintana* DNA and was subsequently found to be infected with the bacterium despite minimal symptoms [38]. In 2023, an outbreak of solid-organ-donor-derived *B. quintana* was reported in Edmonton, a city that had not previously described local transmission: 6 cases of bacillary angiomatosis due to *B. quintana* were described in solid organ transplant recipients who received organs from three deceased donors with a history of homelessness [17,37,39].

Published and unpublished cases of *B. quintana* are increasing in Canada. The escalation in the percent positivity of *B. quintana* qPCR suggests that this trend is not exclusively due to additional testing. The increased incidence of *B. quintana* disease is reflected in the increase in median IgG titer over time, as elevated titers are correlated with disease severity [8,40]. The reason for the increase in Canadian *B. quintana* cases is unknown but may reflect the increased rates of homelessness, a greater degree of crowding, and the more appropriate testing of individuals with a higher pre-test probability of *B. quintana* infection [41,42].

A discrepancy exists between published cases and data from the NML, as no published cases involved molecular testing at the national reference laboratory. Laboratory data from the NML reveal that Ontario had seven molecularly confirmed cases of *B. quintana*, but none of two published cases from Ontario were associated with the qPCR performed at the NML [35,43]. Five published Manitoba cases were diagnosed by 16S rRNA sequencing at a hospital laboratory and no sample was sent to the NML for a qPCR. Estimating the true burden of *B quintana* in Canada is problematic as most infected individuals have mild symptoms and may not seek care, especially among individuals experiencing homelessness with the associated barriers to healthcare access. This issue is further exacerbated by the fact that the disease is not on Canada’s list of notifiable diseases [28]. Most of the confirmed reports were cases of *B. quintana* endocarditis. As endocarditis is deemed to be a rare and severe complication of *B. quintana* infection, affecting less than 20% of those with bacteremia, the elevated number of Canadian endocarditis cases suggests that many cases of *B. quintana* infection remain undiagnosed [1,40,44].

The 19% mortality rate of confirmed *B. quintana* cases from the Canadian literature is similar to the mortality rates of *B. quintana* endocarditis globally and likely reflects the disproportionate number of endocarditis cases compared to other manifestations of *B. quintana* [11]. This mortality rate is comparable to the unpublished description of four cases of *B. quintana* endocarditis occurring in Ottawa in 2022–2023 where one individual died [45].

Our results expose the inadequacies of the current diagnostic tests for *B. quintana.* PCR on whole blood samples lacks sensitivity, even in cases of endocarditis. Elevated blood Ct values may be mistaken for a negative result, leading to misdiagnoses and possible poor outcomes due to inappropriate treatment. This suggests that elevated Ct values (e.g., >35) on whole blood specimens should be further investigated. The serology is limited both by a lack of sensitivity and specificity. The immunologic cross-reactivity of *B. quintana* IgG to *B. henselae* antigens is well-described, and, thus, IFA cannot be used to speciate cases of *Bartonella* [46,47]. Comparing titers to *B. quintana* and *B. henselae* antigens does not solve the issue. Of 11 molecularly proven cases of *B. quintana* with a paired IFA, two had higher titers to the *B. henselae* antigen. Some reference laboratories use a two-fold difference in IFA titer between *B. quintana* and *B. henselae* antigens to denote the causative species. The data presented here show that this protocol may lead to misdiagnoses. Furthermore, serologic positivity may persist for months to years after a cure (especially in cases of endocarditis). Three cases of bacillary angiomatosis and one case of *B. quintana* DNA detection in blood were associated with a negative serology. These false-negative results highlight the limitations of IFA testing as a screening tool. Recent studies have used next-generation sequencing techniques such as microbial cell-free DNA sequencing to diagnose *B. quintana,* although further research is needed to determine how these techniques compare to other diagnostic tools [48,49].

This retrospective laboratory study and systematic review are subject to several limitations. Statistical analyses were applied post hoc, increasing the chance of a false discovery. Laboratory data were devoid of clinical and epidemiologic information, other than tissue type, and year and province of sample acquisition. The possibility of unpublished *B. quintana* cases diagnosed by a qPCR performed outside the NML suggests that our total number of confirmed cases may be an under-estimate. The systematic review may be influenced by bias present in the original studies as well as a publication bias.

## 5. Conclusions

This study reveals that *B. quintana* is a fatal disease with a poorly described epidemiology across many different Canadian jurisdictions. The recent increase in Canadian cases of *B. quintana* signals the need for a coordinated public health response. A few measures would greatly improve our ability to mitigate *B. quintana* transmission in Canada. Including *B. quintana* on Canada’s list of national notifiable diseases would enable centralized data analysis and facilitate the monitoring of epidemiologic trends and the response to future public health interventions [28]. Data acquisition may be facilitated via the mandatory laboratory reporting of PCR-confirmed *B. quintana* cases, as occurs with other diseases of public health concern [28,50]. Laboratory reporting may then be linked to standardized case definitions and outbreak investigation protocols. Such protocols should prioritize screening close contacts of index *B. quintana* cases to identify subclinical disease in communities with known transmission. Due to the limited sensitivity of *B. quintana* PCR on blood samples and the limited specificity of *Bartonella* serology, louse surveillance studies provide an additional tool with which to elucidate the *B. quintana* epidemiology, as the bacterium replicates to high concentrations in the louse gut [51]. Vector surveillance programs already exist for certain tick-borne and mosquito-borne diseases in Canada [52,53]. These programs may be expanded to include *B. quintana,* as well as other louse-borne diseases such as *Rickettsia prowazekii* (epidemic typhus) and *Borrelia recurrentis* (louse-borne relapsing fever). Lastly, *B. quintana* awareness campaigns targeting healthcare practitioners may facilitate early *B. quintana* diagnosis prior to the development of endovascular complications.

## Figures and Tables

**Figure 1 pathogens-13-01071-f001:**
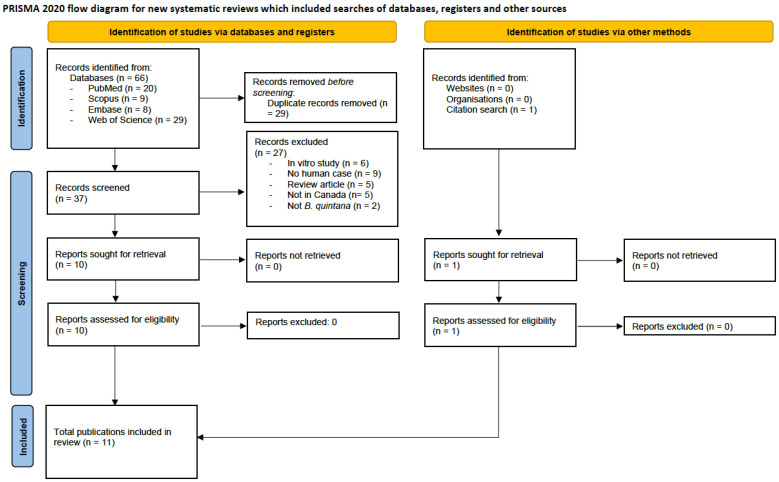
PRISMA 2020 flow diagram for new systematic reviews which included searches of databases, registers, and other sources. Abbreviation: PRISMA, Preferred Reporting Items for Systematic Reviews and Meta-Analyses [36].

**Figure 2 pathogens-13-01071-f002:**
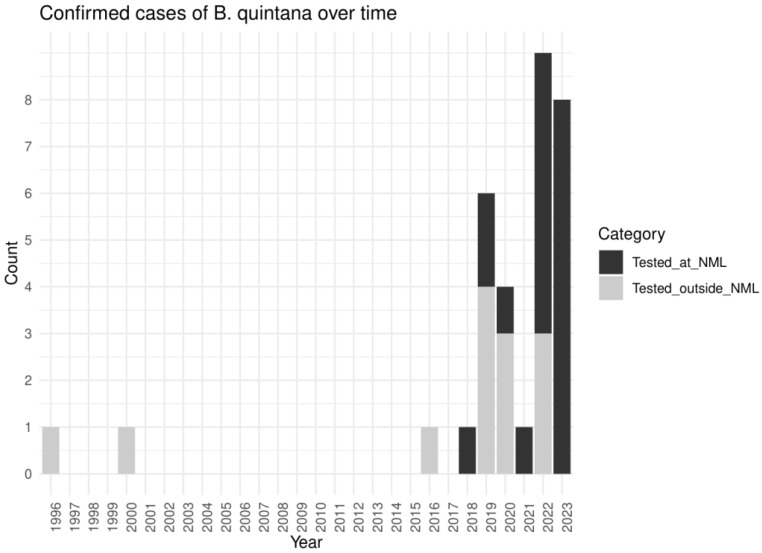
Number of molecularly confirmed cases of *B. quintana* in Canada per year and source of case description (National Microbiology Laboratory data vs. systematic review of published case). *B. quintana* is increasingly diagnosed in Canada with most cases diagnosed in the last five years.

**Figure 3 pathogens-13-01071-f003:**
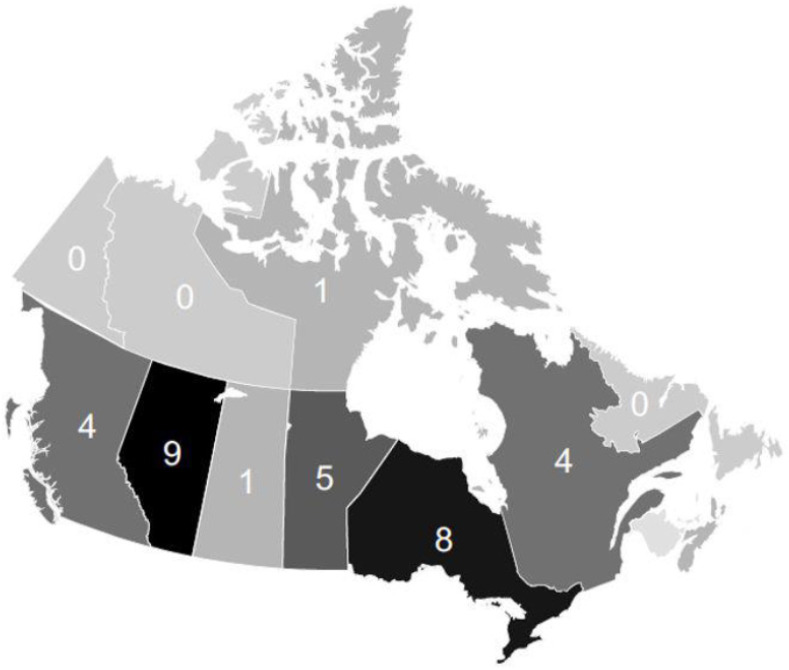
Map of Canada with number of *B. quintana* cases confirmed by PCR per province/territory. *B. quintana* has been acquired in seven provinces and one territory. The provinces with the highest number of reported cases are Alberta (9) and Ontario (8).

**Table 1 pathogens-13-01071-t001:** qPCR detection of *B. quintana* at the NML over time.

Year	N *B. quintana*	Tests Performed	% Positivity	95% CI
2018	1	229	0.44%	0.01–2.41%
2019	2	283	0.71%	0.09–2.53%
2020	2	251	0.80%	0.10–2.85%
2021	1	389	0.26%	0.01–1.42%
2022	8	341	2.35%	1.02–4.57%
2023	12	348	3.45%	1.79–5.95%
Total	26	1841	1.41%	0.87–1.95%

N *B. quintana*: number of samples that tested positive for *B. quintana* DNA by qPCR performed at the NML. Tests performed: number of *Bartonella* qPCR tests performed by the NML (qPCR testing for *Bartonella* at the NML includes testing for both *B. quintana* and *B. henselae*). % positivity: (N *B. quintana*)/(tests performed). 95% CI: 95% confidence interval (binomial “exact” calculation).

**Table 2 pathogens-13-01071-t002:** *B. quintana* cases diagnosed at the NML and associated province, cycle threshold values, and specimen type.

Sample	Case	Year	Province	*Bartonella* Screen Ct	*B. quintana* Ct	Specimen Type
1	1	2018	AB	20.0	20.0	Cardiac valve (mitral)
2	2	2019	BC	15.5	15.1	Cardiac valve (aortic)
3	3	2019	AB	37.5	36.4	Serum
4	4	2020	BC	37.1	36.5	Whole blood
5	4	2020	BC	16.0	16.9	Cardiac valve (valve not specified)
6	5	2021	AB	20.3	18.9	Cardiac valve (aortic)
7	6	2022	ON	39.7	36.9	Skin (face)
8	7	2022	AB	25.7	23.1	Skin (arm)
9	8	2022	BC	21.9	19.7	Cardiac valve (valve not specified)
10	9	2022	ON	18.5	15.4	Cardiac valve (aortic)
11	10	2022	AB	38.4	37.5	Whole blood
12	10	2022	AB	23.1	19.7	Skin (back)
13	11	2022	ON	26.6	23.0	Cardiac valve (valve not specified)
14	11	2022	ON	25.5	22.0	Cardiac valve (valve not specified)
15	12	2023	QC	39.3	35.1	Whole blood
16	12	2023	QC	16.4	13.9	Cardiac valve (valve not specified)
17	13	2023	AB	35.3	32.3	Whole blood
18	13	2023	AB	21.3	18.6	Skin (back)
19	14	2023	BC	25.6	23.1	Skin (arm)
20	15	2023	ON	38.5	36.2	Whole blood
21	16	2023	ON	13.2	11.2	Cardiac valve (aortic)
22	16	2023	ON	15.3	13.3	Cardiac valve (mitral)
23	17	2023	AB	24.4	22.0	Skin (knee)
24	18	2023	ON	29.2	24.1	Cardiac valve (aortic)
25	19	2023	ON	32.2	28.7	Cardiac valve (mitral)
26	19	2023	ON	28.6	25.0	Cardiac valve (mitral)

Case: Patient diagnosed with *B. quintana* disease by PCR performed at the NML. Seven cases were associated with 2 specimens. AB: Alberta. BC: British Columbia. QC: Québec. ON: Ontario. *Bartonella* Screen Ct: Cycle threshold value of ITS3 gene targeting the *Bartonella* genus. *B. quintana* Ct: cycle threshold value of *yopP* gene targeting *B. quintana.*

**Table 3 pathogens-13-01071-t003:** IFA testing at the NML per year and median titers.

Year	N +	N Performed	% Pos	95% CI	Median Titer	IQR
2008	64	653	9.80%	7.63–12.34%	1:64	64
2009	103	787	13.09%	10.81–15.65%	1:64	64
2010	39	796	4.90%	3.51–6.64%	1:64	64
2011	74	1272	5.82%	4.60–7.25%	1:64	64
2012	63	914	6.89%	5.34–8.73%	1:64	64
2013	56	1044	5.36%	4.08–6.91%	1:64	64
2014	73	1055	6.92%	5.46–8.62%	1:64	64
2015	60	1245	4.82%	3.70–6.16%	1:64	64
2016	37	1297	2.85%	2.02–3.91%	1:64	128
2017	51	1479	3.45%	2.55–4.46%	1:128	192
2018	74	1561	4.74%	3.74–5.92%	1:128	64
2019	46	1636	2.81%	2.07–3.73%	1:128	192
2020	44	1555	2.83%	2.06–3.78%	1:128	960
2021	46	1775	2.59%	1.90–3.44%	1:128	960
2022	56	1532	3.66%	2.77–4.72%	1:128	320
2023	76	1903	3.99%	3.16–4.97%	1:128	192
Total	962	20,504	4.69%	4.41–4.99%	1:64	64

N +: number of IFA tests that were reported as positive for the *B. quintana* antigen. N performed: number of IFA tests performed at the NML. % pos: N +/N ordered. 95% CI: 95% confidence interval (binomial “exact” method). IQR: interquartile range of IFA titer.

**Table 4 pathogens-13-01071-t004:** *B. quintana* qPCR positive cases with paired IFA results.

Case	*Bartonella* Ct	Bq Ct	Specimen Type	Bq Titer	Bh Titer
1	20.0	20.0	Cardiac valve (mitral)	1:4096	1:4096
2	37.5	36.4	Serum	1:1024	1:1024
3	20.3	18.9	Cardiac valve (aortic)	1:2048	1:2049
4	38.4	37.5	Blood	<1:64	<1:64
5	35.3	32.3	Blood	1:64	1:256
6	25.7	23.1	Skin (arm)	<1:64	<1:64
7	23.1	19.7	Skin (back)	<1:64	<1:64
8	39.7	36.9	Skin (face)	1:4096	1:4096
9	21.3	18.6	Skin (unknown)	1:64	1:64
10	25.6	23.1	Skin (arm)	1:512	1:4096
11	24.4	22.0	Skin (knee)	1:256	1:128

*Bartonella* Ct: PCR cycle threshold value for the ITS3 gene targeting the *Bartonella* genus, used as a molecular screening test. Bq Ct: PCR cycle threshold value for the *yopP* gene targeting *B. quintana* species. Bq titer: IFA IgG titer result to the *B. quintana* antigen. Bh titer: IFA IgG titer result to the *B. henselae* antigen.

## Data Availability

Additional data are available in the Appendix A.

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
