# Peer review of "Bartonella quintana* Infection in Canada: A Retrospective Laboratory Study and Systematic Review of the Literature"

_pathogens, 2024, doi:10.3390/pathogens13121071_

Round 1

Reviewer 1 Report

Comments and Suggestions for Authors

Thank you for the review opportunity.

I find the article by Boodman et. al a good read, worth considering for publication.

I do have some minor suggestions:

"The detection of B. quintana disease in seven provinces and one territory suggests that B. quintana has a national distribution. B. quintana disease is increasingly diagnosed in Canada, sug- gesting ongoing transmission across geographic settings. " A little too much suggesting, please rephrase :)

Before the epidemiology in Canada paragraph, i would add a short paragraph about the epidemiology of Bartonella in the world, please consider.

Please finish the introduction section with a paragraph with the reasoning and aims of your work .

"immunoglobulin G (IgG) antibodies" immunoglobulins and antibodies are synonyms, please delete antibodies

Otherwise, the Materials and Methods are really well written.

Results

I would include a map figure with Canada detailing how many cases were found per region. It would be a nice addition to your work.

Figure 1 is not clear for me. Please consider up-ing the resolution.

Ah, now i've noticed Figure 3. Please consider moving this Figure at the beginning of the Results section.

conclusion -> Conclusion

Please consider expanding the Conclusions section, what can your data be used for ? prevention programs ? screening in specific areas as areas with  high homelessness.

Besides my minor suggestions, i consider this article should be considered for publication with minor revisions.

Wishing you the best,

Reviewer

Author Response

Thank you for the review opportunity.

I find the article by Boodman et. al a good read, worth considering for publication.

Response: Thank you for your comments. Your review strengthened our manuscript.

I do have some minor suggestions:

"The detection of B. quintana disease in seven provinces and one territory suggests that B. quintana has a national distribution. B. quintana disease is increasingly diagnosed in Canada, sug- gesting ongoing transmission across geographic settings. " A little too much suggesting, please rephrase :)

Response: Thank you. We agree that increased B. quintana diagnosis “indicates” ongoing transmission. We have modified the abstract accordingly (please see lines 38-39).

Before the epidemiology in Canada paragraph, i would add a short paragraph about the epidemiology of Bartonella in the world, please consider.

Response: Thank you. We agree and have added a short paragraph on the global epidemiology of Bartonella quintana as suggested (please see lines 62-71). As our paper focuses on B. quintana, we discussed the global epidemiology of this species, rather than attempt to discuss the epidemiology of the 40+ different Bartonella species, as each species has its own distinct epidemiology.

Please finish the introduction section with a paragraph with the reasoning and aims of your work .

Response: Thank you. We agree. The current last paragraph of the introduction articulates the reasoning (e.g., not notifiable disease, epidemiology poorly described) and the aims (e.g., comprehensive description of Canadian B. quintana epidemiology) of the work (please see lines 82-90).

"immunoglobulin G (IgG) antibodies" immunoglobulins and antibodies are synonyms, please delete antibodies

Response: Thank you. We have corrected this (please see line 107).

Otherwise, the Materials and Methods are really well written.

Response: Thank you.

Results

I would include a map figure with Canada detailing how many cases were found per region. It would be a nice addition to your work.

Response: Thank you. This has been done (see figure 3).

Figure 1 is not clear for me. Please consider up-ing the resolution.

Response: Thank you for bringing this to our attention. We have increased the resolution.

Ah, now i've noticed Figure 3. Please consider moving this Figure at the beginning of the Results section.

Response: Thank you. We are glad that you appreciate figure 3. The journal states that figure placement must occur after its first use in the text. Since figure 3 combines data from both the National Microbiology Laboratory and the systematic review, it cannot be placed higher up.

conclusion -> Conclusion

Response: Thank you. This has been corrected.

Please consider expanding the Conclusions section, what can your data be used for ? prevention programs ? screening in specific areas as areas with  high homelessness.

Response: Thank you. We have expanded the conclusion section, as suggested (please see lines 515-534).

Besides my minor suggestions, i consider this article should be considered for publication with minor revisions.

Wishing you the best.

Response: Thank you very much for your helpful review.

Reviewer 2 Report

Comments and Suggestions for Authors

Thank you for the opportunity to read this manuscript. The authors performed an analysis of laboratory data from Canada’s 24 National Microbiology Laboratory (NML) and combined in with a systematic review of the literature, in an effort to understand respective Bartonella epidemiology in Canada.

Overall I think it is a valuable manuscript however, I have a few suggestions that could improve this report.

·       A deeper comparison with other cohorts in other territories would be definitely of use. Please see relative literature  e.g Shepard et al 2022 Journal of Infectious Diseases; Drancourt et al 1995 New England Journal of Medicine etc

·       vector-borne pathogens that affect similar populations (e.g., Borrelia spp., Rickettsia spp.) could also broaden the discussion. Consider discussing how socioeconomic factors associated with B. quintana transmission (e.g., houselessness, lack of sanitation) compare to other vector-borne diseases that disproportionately affect similar demographics. Especially of interest would be the discussion of data from other high income coutnries. This would provide a more comprehensive view of B. quintana in the larger context of infectious diseases.

    • Discussing the limitations of diagnostic techniques, such as PCR and serology, for B. quintana relative to other vector-borne diseases (e.g., Rickettsia spp. often rely on serologic confirmation) could further validate the selected approach. In discussing diagnostic challenges, such as PCR sensitivity and IFA limitations, expand on the potential for false negatives. Comparisons with alternative diagnostic methods, such as next-generation sequencing or MALDI-TOF MS, which are emerging in vector-borne disease diagnostics might be useful.

·       Certain sentences in the abstract are complex, e.g, "Canadian publications describe B. quintana endocarditis among populations experiencing houselessness and Indigenous communities without access to running water." Could be simplified to è "B. quintana endocarditis has been reported primarily in populations experiencing homelessness and in Indigenous communities with limited access to water."

·       Ensure consistent use of terms like “houselessness” versus “homelessness” and “pediculosis corporis” versus “body lice infestation.” Ensure overall cohesiveness.

·       While the conclusion hints at a need for a coordinated public health response, please emphasize by suggesting specific actions, e.g national reporting requirements, public health awareness programs, improved access to diagnostic services for at-risk populations etc.

·       Figures and tables are helpful but could be supplemented with brief summaries to aid readability e.g include a small note about key trends alongside Figure 2 and Figure 3

Author Response

Reviewer 2

Thank you for the opportunity to read this manuscript. The authors performed an analysis of laboratory data from Canada’s 24 National Microbiology Laboratory (NML) and combined in with a systematic review of the literature, in an effort to understand respective Bartonella epidemiology in Canada.

Overall I think it is a valuable manuscript however, I have a few suggestions that could improve this report.

Response: Thank you for your reviews which improve our manuscript.

  • A deeper comparison with other cohorts in other territories would be definitely of use. Please see relative literature  e.g Shepard et al 2022 Journal of Infectious Diseases; Drancourt et al 1995 New England Journal of Medicine etc

Response: Thank you. We agree and have included descriptions of other cohorts, specifically Shepard et al. 2022 and Drancourt et al. 1995, as suggested (please see lines 61-75). We described these studies in the introduction to consolidate your review with that of the other reviewer.

  • vector-borne pathogens that affect similar populations (e.g., Borreliaspp., Rickettsia spp.) could also broaden the discussion. Consider discussing how socioeconomic factors associated with B. quintana transmission (e.g., houselessness, lack of sanitation) compare to other vector-borne diseases that disproportionately affect similar demographics. Especially of interest would be the discussion of data from other high income coutnries. This would provide a more comprehensive view of B. quintana in the larger context of infectious diseases.

Response: Thank you. We have included a brief description of Rickettsia prowazekii and Borrelia recurrentis in the final section proposing that arthropod surveillance programs include louse-borne disease (please see lines 375-377). However, as this article focuses on B. quintana in Canada and there have been no confirmed local cases of Rickettsia prowazekii and Borrelia recurrentis in Canada, we have opted to avoid describing these diseases in detail.

Discussing the limitations of diagnostic techniques, such as PCR and serology, for B. quintana relative to other vector-borne diseases (e.g., Rickettsia spp. often rely on serologic confirmation) could further validate the selected approach. In discussing diagnostic challenges, such as PCR sensitivity and IFA limitations, expand on the potential for false negatives. Comparisons with alternative diagnostic methods, such as next-generation sequencing or MALDI-TOF MS, which are emerging in vector-borne disease diagnostics might be useful.

Response: Thank you. We have discussed the issues of false negativity of both PCR (please see lines 335-338) and IFA (please see lines 346-348). We have added a brief discussion of alternative diagnostic techniques such as NGS, as suggested (please see lines 537-540).

  • Certain sentences in the abstract are complex, e.g, "Canadian publications describe B. quintanaendocarditis among populations experiencing houselessness and Indigenous communities without access to running water." Could be simplified to è "B. quintana endocarditis has been reported primarily in populations experiencing homelessness and in Indigenous communities with limited access to water."

Response: Thank you. We have made the change.

  • Ensure consistent use of terms like “houselessness” versus “homelessness” and “pediculosis corporis” versus “body lice infestation.” Ensure overall cohesiveness.

Response: Thank you. We have consolidated these terms to ensure cohesiveness.

  • While the conclusion hints at a need for a coordinated public health response, please emphasize by suggesting specific actions, e.g national reporting requirements, public health awareness programs, improved access to diagnostic services for at-risk populations etc.

Response: Thank you. This has been done. We have expanded the conclusion, as suggested.

  • Figures and tables are helpful but could be supplemented with brief summaries to aid readability e.g include a small note about key trends alongside Figure 2 and Figure 3.

Response: Thank you. This has been done.

Round 2

Reviewer 2 Report

Comments and Suggestions for Authors

Thank you for revising your manuscript and adopting my suggestions